# Statistical Analysis of the Potential of Landslides Induced by Combination between Rainfall and Earthquakes

Chih-Ming Tseng [1,*], Yie-Ruey Chen [1], Chwen-Ming Chang [2], Ya-Ling Yang [3], Yu-Ru Chen [4] and Shun-Chieh Hsieh [1]

1 Department of Land Management and Development, Chang Jung Christian University, Tainan 711301, Taiwan
2 Department of Business Administration, Chang Jung Christian University, Tainan 711301, Taiwan
3 Department of Aviation and Maritime Transportation Management, Chang Jung Christian University, Tainan 711301, Taiwan
4 Disaster Prevention Research Center, National Cheng Kung University, Tainan 701401, Taiwan
* Correspondence: cmtseng@mail.cjcu.edu.tw

**Abstract:** This study analyzed the potential of landslides induced by the interaction between rainfall and earthquakes. Dapu Township and Alishan Township in Chiayi County, southern Taiwan, were included as study areas. From satellite images and the literature, we collected data for multiple years and time series and then used the random forest data mining algorithm for satellite image interpretation. A hazard index for the interaction between earthquakes and rainfall ($I_{HERI}$) was proposed, and an index for the degree of land disturbance ($I_{DLD}$) was estimated to explore the characteristics of $I_{HERI}$ under specific natural environmental and slope land use conditions. The results revealed that among the investigated disaster-causing factors, the degree of slope land use disturbance, the slope of the natural environment, and rainfall exerted the strongest effect on landslide occurrence. When $I_{HERI}$ or $I_{DLD}$ was higher, the probability of a landslide also increased, and under conditions of a similar $I_{DLD}$, the probability of landslides increased as the $I_{HERI}$ value increased, and vice versa. Thus, given the interaction between rainfall and earthquakes in the study area, the effect of the degree of slope land use disturbance on landslides should not be ignored. The results of a receiver operating characteristic (ROC) curve analysis indicated that the areas under the ROC curve for landslides induced by different trigger factors were all above 0.94. The results indicate that the area in which medium–high-level landslides are induced by an interaction between rainfall and earthquakes is large.

**Keywords:** rainfall; earthquake; satellite image interpretation; landslide potential; random forest; geographic information system



## 1. Introduction

Taiwan is an island located in an earthquake zone. Taiwan frequently experiences various natural disasters, such as typhoons and earthquakes. Furthermore, global climate change has adversely affected Taiwan in recent years by causing frequent extreme rainfall, short-term heavy rainfall, and disturbances in slope land use; these factors can result in landslides and substantial changes in mountainous areas. Therefore, studies should investigate the relationship between landslides caused by the interaction of factors that trigger rainfall and earthquakes, particularly those related to the target region's natural environment and disturbances in slope land use, and the scale of landslides. The index of the interaction effect of the dual trigger factors and the degree of disturbance in slope land use should be determined; they can serve as preliminary standards that can be used to effectively regulate the degree of slope land use and development and thus prevent slope land disasters.

Some scholars have used a genetic adaptive neural network and performed texture analysis to interpret satellite images and investigate the relationship between land

use and rainfall-induced landslides; these approaches can be beneficial for landslide monitoring [1–3]. Feng et al. [4] reported that the combination of a random forest (RF) with texture analysis exhibited a higher accuracy in mapping land covered by vegetation in cities than did the traditional maximum likelihood method. Stumpf et al. [5] indicated that the interpretation and classification of satellite images can be helpful for analyzing big data, obtaining cadastral information, and determining land cover and vegetation types and soil properties.

Zhong [6] used the 1999 Chi-Chi earthquake as a boundary and selected two typhoons before and after the earthquake to investigate the effect of the interaction between the earthquake and rainfall on slope stability. They determined that before and after the earthquake, the number and areas of landslides were positively correlated with the maximum rainfall intensity and the slope of a landslide caused by higher rainfall was lower than that of a landslide caused by lower rainfall. In addition, Lin et al. [7] evaluated the impact of the Chi-Chi earthquake on subsequent rainfall-induced landslides and determined that the rainfall-induced landslides were mostly distributed between 40° and 50° and between 30° and 40° after and before the earthquake, respectively. Tang et al. [8] reported that after the Wenchuan earthquake, subsequent torrential rain caused a 30% increase in landslide areas. Moreover, landslide points were more numerous after the earthquake than they were before the earthquake. Huang et al. [9] suggested that the use of seismic factors can increase the prediction accuracy for landslide occurrence. Chen et al. [10] determined that extreme rainfall events caused frequent landslide erosion, with extreme rainfall (i.e., a maximum 24-h rainfall level of >600 mm) accounting for 64%–79% of the average landslide erosion rate. Yang et al. [11] conducted a sensitivity analysis and employed the analytical hierarchy process and factor weighting to evaluate and compare the occurrence of landslides after and before earthquakes. They determined that landslides caused by strong earthquakes exhibited more spatial clustering. Jan et al. [12] demonstrated that the impact of typhoon rainfall depends not only on the amount of rainfall but also on its intensity. Li et al. [13] observed that new landslides tended to occur in low-altitude or low-slope areas under normal rainfall and in high-altitude areas under high rainfall. Tseng et al. [14] reported that the number and area of landslides induced by rainfall were positively correlated with the degree of land disturbance. Wistuba et al. [15] determined that earthquakes caused up to 50% of landslide events, either alone or in combination with rainfall, but earthquakes were rarely the sole trigger of landslides. Chen et al. [16] reported that the main trigger factors for landslides were cumulative rainfall in areas that were more prone to landslides and rainfall intensity in areas that were less prone to landslides. Extreme climate conditions have caused increases in the magnitude and frequency of landslides in both types of areas that are generally less and more prone to landslides. Valagussa et al. [17] analyzed three earthquake events and discovered that surface disturbance affects the intensity of landslides; the higher the intensity of a landslide is, the greater the surface disturbance is. Moreover, they proposed that in addition to surface disturbance, terrain type and lithology affect the intensity of landslides. Quesada-Román et al. [18] investigated the combined effects of earthquakes and rainfall on landslides and debris flow. They observed that the density of landslides was higher in areas closer to earthquake epicenters and that more rainfall and landslides occurred in areas with higher slopes and elevations. Ruggeri et al. [19] demonstrated the possibility to distinguish the difference between seismic and rainfall induced displacements of the slope. Bontemps et al. [20] reported that moderate-intensity earthquakes with a magnitude of >5.5 on the Richter scale increased the probability of landslide occurrence and rainfall events either before or after the earthquake and therefore led to more severe landslides. A combination of low-intensity earthquakes with a magnitude of 3.2–3.6 on the Richter scale and heavy rainfall reduced the stability of slopes. Liu et al. [21] analyzed the debris flow records of the Wenchuan, Lushan, and Jiuzhaigou earthquakes and determined that the critical values of postearthquake landslides were smaller than those of pre-earthquake rainfall.

Artificial intelligence and decision trees are commonly used for data mining when big data are analyzed. Breiman et al. [22] reported that an RF is a combination of decision tree classifiers; each tree is influenced by independently sampled random vector values, and the distribution is the same for all decision trees in a forest. Colditz et al. [23] reported that the RF algorithm yielded the best results for the area ratio assignment of the training samples for each class. Belgiu et al. [24] demonstrated that an RF is a classifier that aggregates several decision trees and can handle high data dimensionality and multiplicity. Lagomarsino et al. [25] Taalab et al. [26] reported that an RF can process a large amount of data. In earth science and landslide sensitivity research, an RF is considered an advanced and mature technology. Catani et al. [27] demonstrated that the dimension of parameter space, the mapping unit and the training process strongly influence the classification accuracy and the prediction results. Goetz et al. [28] presented a comparison of traditional statistical and novel machine learning models applied for landslide susceptibility modeling. Steger et al. [29] explored discrepancies between the predictive performance of a landslide susceptibility model and the geomorphic plausibility of subsequent landslide susceptibility maps. Chen et al. [30] used kernel logistic regression, an RF, and an alternating decision tree to build a potential map of groundwater sources. They observed the highest area under the curve (AUC) value and degree of accuracy for the RF.

Ercanoglu [31] used a neural network in combination with a geographic information system (GIS) to draw a potential map of slope land hazards on the basis of landslide locations identified on aerial photos and six parameters, namely slope, aspect, elevation, terrain, a humidity index, and a vegetation coverage index. Shahabi et al. [32] combined remote sensing and GISs to apply statistical models for the delineation of landslide-sensitive areas. They considered the following factors: slope, aspect, elevation, lithology, normalized difference vegetation index, vegetation, rainfall, distance from the fault, distance from a water system, and distance from a road. Tseng et al. [14] employed the time range before and after a typhoon that had recently caused a road slope landslide in the study area as a benchmark and interpreted images to determine surface changes before and after the occurrence of landslides to establish a landslide potential assessment model. Then, they used a GIS platform to plot a landslide potential map.

The present study used an RF algorithm to interpret satellite images, quantitatively analyzed landslide potential and hazard factors induced by the interaction between rainfall and earthquakes, and explored the characteristics of landslides induced by the interaction between rainfall and earthquakes in a specific natural environment under slope land utilization and development conditions in the region.

## 2. Research Methods

### 2.1. RF

An RF is a multifunctional machine learning algorithm. The algorithm contains numerous decision tree classifiers without any relationships. An RF can handle a large amount of input data, smooth out errors in class determination, and maintain accuracy by classifying data when data are insufficient or missing [33]. An RF selects a subset to predict the growth of a decision tree and determine the growth of each tree in accordance with the bootstrap training set. The bootstrapping method involves repeatedly sampling a limited sample with replacement to establish a new sample that can adequately represent the distribution of the population [34,35]. The randomly sampled bootstrap samples and randomly selected input factors are used to construct multiple CARTs (Classification and Regression Tree) in accordance with the maximum depth, and generate forests with the lowest generalization error. Each CART in the random forest grows independently and uses different random samples of bootstraps. Therefore, each CART has no relationship. After the establishment of the random forest, when a new sample is input, each decision tree in the forest judges separately and the result is dependent on majority voting. When creating a random forest, there is no over-fitting problem [22,35,36]. The RF method is widely used in different fields and has provided satisfactory results in studies analyzing

the potential of landslide occurrence [37,38]. In addition, the RF method is suitable for data with many variables or insufficient resolution, and it does not lead to the problem of overfitting [22].

In the present study, we used the Train Random Trees Classifier (ESRI, 2021) in ArcGIS (10.5.1) to interpret and classify satellite images, and the RF module [39] developed in the R language was used to establish a landslide potential assessment model.

## 2.2. Texture Analysis

Texture is an image characteristic, and it can be analyzed to distinguish groups of images. Mathematically, texture is defined as the correlation of the grayscale value or color space of adjacent pixels in an image or the visual representation of image grayscale values and color changes with spatial positions, including edges, shapes, stripes, and color blocks [40]. This study used the image processing software ERDAS IMAGINE to analyze image texture. Four aspects of textures can be analyzed using ERDAS IMAGINE, namely variance, skewness, kurtosis, and mean Euclidean distance [41].

## 2.3. Accuracy Assessment

To determine whether the accuracy of the satellite image interpretation results met the requirement for land use classification data, we adopted the error matrix method and evaluated the accuracy of the image classification. After the error matrix analysis and calculation are completed, several commonly used accuracy evaluation indicators can be calculated, including producer's accuracy (PA), user's accuracy, overall accuracy (OA), and coefficient of agreement (Kappa index) [42,43]. Kappa values are between 0 and 1; the larger the Kappa value is, the higher the classification accuracy is [44].

## 2.4. Receiver Operating Characteristic Curve

A receiver operating characteristic (ROC) curve can be used to describe all the possible combinations of correct and incorrect [45]. The advantage of using an ROC curve is that its value is not affected by the number of classified samples and can be used to evaluate the recognition of patterns. For ROC curves, true positive (*TP*) rates (*TPR*) are used as ordinates and false positive (*FP*) rates (*FPR*) are used as abscissas. To plot an ROC curve, the *TPR*, true negative (*TN*) rate, *FPR*, and false negative (*FN*) rate should be calculated. The *FPR* represents the ratio of all negative samples that are wrongly judged as positive, as presented in Formula (1). The *TPR* represents the ratio of all positive samples that are correctly judged to be positive, as presented in Formula (2).

$$FPR = FP/(FP + TN) \tag{1}$$

$$TPR = TP/(TP + FN) \tag{2}$$

This study used the AUC to determine the advantages and disadvantages of the evaluation model. If the area is assumed to be obtained in the space of $1 \times 1$, the AUC must be between 0 and 1. The higher the AUC value is, the better the discriminative ability is, and vice versa [45].

## 3. Study Areas

This study was conducted in Dapu Township and Alishan Township in Chiayi County, southern Taiwan (Figure 1). Typhoon Morakot occurred in Taiwan in August 2009 and caused record-breaking disasters. The accumulated rainfall in Alishan Township reached 3060 mm and resulted in 619 deaths and 76 missing individuals. At 1:43:3.2 pm on 2 June 2013, an earthquake with a magnitude of 6.5 on the Richter scale and a depth of 14.5 km occurred 29.3 km east of the Nantou County Government headquarters; it killed 4 people and injured 19 people. This earthquake also caused rail bending, rockfalls, and landslides.

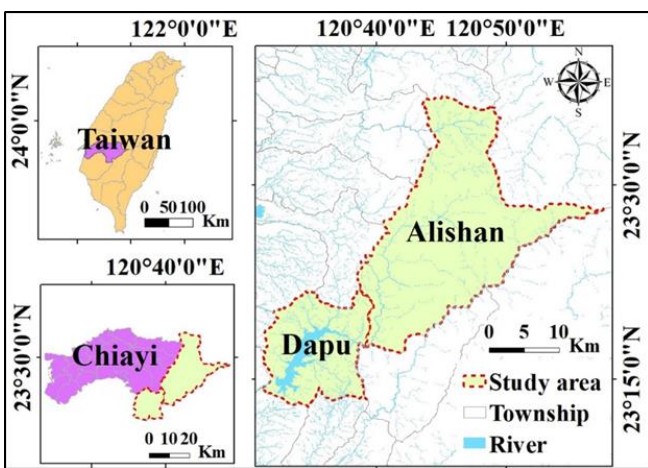

**Figure 1.** Study areas.

Alishan Township is located in the eastern part of Chiayi County, Taiwan. This township has the largest area and lowest population density in Chiayi County. Dapu Township is located in the southeast corner of Chiayi County. This township has the second lowest population density in Chiayi County. Alishan Township and Dapu Township in Chiayi County are located to the north and south of the Tropic of Cancer, respectively. Alishan Township is mainly located in the middle of the Central Mountains. Its altitude varies considerably. The terrain is high in the east and low in the west. The altitude is approximately 360–3997 m. The temperature mostly ranges from 6.3 °C to 15 °C. The average minimum and maximum temperatures are 7.2 °C and 15.5 °C, respectively. Dapu Township is located in the Alishan Mountains, with an altitude of approximately 230–1100 m. An alpine terrain, hills and plains, and a basin are present in the east, west, and center, respectively. The annual average maximum temperature is 28 °C. The annual precipitation in Alishan Township and Dapu Township is approximately 4186 and 2236 mm, respectively [46]. The main river flowing through the study area is Zengwen River, which is the fourth longest stream in Taiwan. Its source is 2609 m above sea level, with a total length of 138.5 km and a drainage area of 1176.7 km$^2$. Its main tributaries are the Tanaku River and Puyanu, Caolan, Houbori, Cailiao, and Guantian streams. Zengwen River is rich in water resources and flows through the Zengwen, Nanhua, and Wushantou reservoirs. This river is also used as a water supply and for power generation and attracts tourists to the local area [47].

## 4. Potential of the Induction of Landslides by the Interaction between Rainfall and Earthquakes

### 4.1. Interpretation and Classification of Satellite Images before and after Rainfall or Earthquakes in the Study Area and Extraction of Landslide Data

In this study, we collected Formosa Satellite 2, SPOT-5, SPOT-6, and SPOT-7 images with image resolutions of 6 × 6 M, 8 × 8 M, and 10 × 10 M, respectively. We used ERDAS IMAGINE to fuse various images into a multispectral image and then performed image positioning. Subsequently, the editor function of ArcGIS was used to delineate the parts covered by clouds or shadows in the study area and then crop and remove the clouds. To ensure the integrity of the satellite images before and after each rainfall or typhoon in the study area and reduce the impact of cloud cover, we first distinguished between the time periods before and after each occurrence of rainfall or each earthquake. For each time period, if a preprocessed image had a blank block because of cloud removal, the Mosaic to New Raster function in ArcGIS (10.5.1) was used to map and merge preprocessed satellite images belonging to the same time period and with similar image dates to obtain a satellite image with low or no cloud coverage. According to the method reported by Chen et al. [10], we used ERDAS IMAGES [41] to analyze the texture of the aforementioned multispectral images to improve the accuracy of interpretation and classification.

With reference to Chen et al. [1], we used seven classifications, namely water, roads, buildings, streamways, bare ground, green cover, and cash crops (fruit trees and tea plants), as interpretive classifications. The ArcGIS image classification module (segmentation and classification toolset) in the random tree classifier (train random trees) was used for the classification and interpretation of the satellite images. To confirm the accuracy of the collected satellite images after interpretation, we used the aforementioned coefficient of agreement (Kappa) and OA as the bases for accuracy evaluation and conducted random sampling of the selected images. Seven interpretation classifications were determined, and 25 points (interpretation grids) were randomly selected as checkpoints. High-resolution aerial photos and on-site survey data were used for cross comparison. The Kappa values and OA of the satellite image interpretation and classification results before and after different rainfall and earthquake events in the study area are summarized in Table 1. The average Kappa value and OA of all 37 satellite image interpretation results were 0.66 and 70.1%, respectively, and the overall image classification accuracy was medium to medium high.

**Table 1.** Coefficient of agreement and overall accuracy of satellite image interpretation results.

| Satellite Image | | | | | *Kappa* | *OA* (%) | *Kappa* (Mosaic) | *OA* (%) (Mosaic) |
|---|---|---|---|---|---|---|---|---|
| **No** | **Year** | **Date** | **Before/After Event** | **Township** | | | | |
| 1 | 2004 | 10 Feb. | Before Typhoon Mindulle | Alishan | 0.60 | 65.7 | | |
| 2 | 2004 | 10 July | After Typhoon Mindulle | Alishan | 0.75 | 77.1 | | |
| 3 | 2008 | 5 Jan. | Before the 0305 Earthquake and Typhoon Kalmaegi | Alishan | 0.65 | 68.0 | 0.65 | 67.5 |
| 4 | 2008 | 10 Jan. | Before the 0305 Earthquake and Typhoon Kalmaegi | Alishan | 0.64 | 66.9 | | |
| 5 | 2008 | 21 July | After the 0305 Earthquake and Typhoon Kalmaegi | Alishan | 0.75 | 77.1 | | |
| 6 | 2008 | 5 Jan. | Before the 0305 Earthquake and Typhoon Kalmaegi | Dapu | 0.63 | 68.0 | | |
| 7 | 2008 | 21 July | After the 0305 Earthquake and Typhoon Kalmaegi | Dapu | 0.70 | 73.1 | | |
| 8 | 2009 | 12 Apr. | Before Typhoon Morakot and the 1105 earthquake | Dapu | 0.66 | 70.3 | | |
| 9 | 2009 | 6 Nov. | After Typhoon Morakot and the 1105 earthquake | Dapu | 0.62 | 66.9 | | |
| 10 | 2010 | 11 Apr. | Before the 0726 heavy rain | Dapu | 0.67 | 70.9 | | |
| 11 | 2010 | 4 Aug. | After the 0726 heavy rain | Dapu | 0.67 | 71.4 | | |
| 12 | 2010 | 4 Aug. | Before Typhoon Fanapi and the 1108 Earthquake | Dapu | 0.65 | 69.1 | | |
| 13 | 2010 | 27 Dec. | After Typhoon Fanapi and the 1108 Earthquake | Dapu | 0.65 | 69.1 | | |
| 14 | 2011 | 27 July | Before the Typhoon Nanmadol | Dapu | 0.71 | 74.9 | 0.68 | 72.0 |
| 15 | 2011 | 16 Aug. | Before the Typhoon Nanmadol | Dapu | 0.64 | 69.1 | | |
| 16 | 2011 | 27 Sep. | After the Typhoon Nanmadol | Dapu | 0.61 | 66.9 | | |
| 17 | 2012 | 10 Feb. | Before the 0226 earthquake | Dapu | 0.78 | 80.0 | | |
| 18 | 2012 | 7 Mar. | After the 0226 earthquake | Dapu | 0.66 | 70.2 | | |
| 19 | 2013 | 2 June | Before the 0602 earthquake | Alishan | 0.68 | 71.4 | | |

**Table 1.** *Cont.*

| | | Satellite Image | | | *Kappa* | *OA* (%) | *Kappa* (Mosaic) | *OA* (%) (Mosaic) |
|---|---|---|---|---|---|---|---|---|
| No | Year | Date | Before/After Event | Township | | | | |
| 20 | 2013 | 29 June | After the 0602 earthquake | Alishan | 0.62 | 65.7 | 0.63 | 66.3 |
| 21 | 2013 | 4 July | After the 0602 earthquake | Alishan | 0.63 | 66.9 | | |
| 22 | 2015 | 28 Feb. | Before the 0520 heavy rain | Alishan | 0.66 | 68.6 | | |
| 23 | 2015 | 10 June | After the 0520 heavy rain | Alishan | 0.68 | 70.9 | | |
| 24 | 2015 | 5 Mar. | Before the 0520 heavy rain | Dapu | 0.73 | 74.9 | | |
| 25 | 2015 | 10 June | After the 0520 heavy rain | Dapu | 0.70 | 73.1 | | |
| 26 | 2015 | 28 Nov. | Before the 0206 earthquake | Alishan | 0.61 | 65.7 | | |
| 27 | 2016 | 14 Feb. | After the 0206 earthquake | Alishan | 0.64 | 66.9 | 0.68 | 70.3 |
| 28 | 2016 | 30 Mar. | After the 0206 earthquake | Alishan | 0.72 | 73.7 | | |
| 29 | 2015 | 28 Nov. | Before the 0206 earthquake | Dapu | 0.63 | 68.0 | | |
| 30 | 2016 | 30 Mar. | After the 0206 earthquake | Dapu | 0.66 | 70.3 | | |
| 31 | 2016 | 30 Mar. | Before Typhoon Megi | Dapu | 0.66 | 69.7 | | |
| 32 | 2016 | 19 Nov. | After Typhoon Megi | Dapu | 0.64 | 68.6 | | |
| 33 | 2017 | 18 Oct. | Before the 1122 earthquake | Alishan | 0.60 | 64.6 | 0.62 | 66.3 |
| 34 | 2017 | 17 Nov. | Before the 1122 earthquake | Alishan | 0.64 | 68.0 | | |
| 35 | 2018 | 16 Jan. | After the 1122 earthquake | Alishan | 0.66 | 69.7 | | |
| 36 | 2017 | 17 Nov. | Before the 0320 earthquake | Dapu | 0.65 | 68.6 | | |
| 37 | 2018 | 9 Apr. | After the 0320 earthquake | Dapu | 0.72 | 74.3 | | |
| | Total Average | | | *Kappa* = 0.66 *OA* = 70.1% | | | | |

To accurately determine the locations of landslide areas in the study area, the images of the aforementioned rainfall and earthquake events were used to manually identify the locations of exposed areas. The bare ground in the images before and after an earthquake event was subtracted to determine the location of the landslide area. Among the types of slope failure, debris slides were the easiest and most reliable type to be identified on satellite images since vegetation was effectively stripped off from the slopes. Therefore, debris slides are the major landslides mapped in our study.

### 4.2. Selection of Landslide Hazard Factors

This study analyzed three environmental hazards that can induce landslides, namely the natural environment, slope land use disturbance, and trigger factors. Elevation, slope, slope aspect, geology, distance from the water system, and distance from the fault were initially selected as natural environmental hazards. Road density, building density, crop density, green coverage, and bare land density were selected as hazard factors for slope land use disturbance. Rainfall and earthquake events were included as trigger factors. The product of effective accumulated rainfall (EAR) and the maximum 3-h rolling rainfall intensity ($I_{3Rmax}$) were used as the rainfall indicator. Peak ground acceleration (PGA) was used as the earthquake indicator. The methods for estimating and grading the various hazard factors were in accordance with those reported by Tseng et al. [14] and are explained in the following.

4.2.1. Natural Environmental Factors

A.　Elevation

We used ArcGIS Spatial Analyst to determine the digital elevation mode (DEM) in the study area and ArcGIS Zonal Statistics to estimate the average elevation value in each basic grid of the study area. The elevation was divided into seven grades, with an interval of 250 m, and then graded and coded. The smaller the code was, the higher the elevation value was.

B.　Slope

We used the DEM of the study area and the slope analysis tool (Slope) in ArcGIS to estimate the average slope value of each basic grid and divided the slope into six grades, with an interval of 10%. The sixth grade slope was coded as 1, and the first grade slope was coded as 6. The smaller the code was, the greater the slope was.

C.　Aspect

According to the method reported by Tseng et al. [14], we used the ArcGIS aspect analysis tool (Aspect) to estimate the average aspect value for each basic grid. Because the slope aspect of the land with exposed mudstone in southwestern Taiwan mainly faces south, south was coded as 1 and southeast and southwest were coded as 2. The aspects were divided into five grades, and the flat land was coded as 0.

D.　Distance from the river

According to Moayedi et al. [48], an area more than 700 m away from a river is less likely to experience a landslide. Hong et al. [49] classified the distance of an area from a river into the levels of 100 m or less, 100–300 m, 300–500 m, 500–700 m, and above 700 m. Therefore, to match the distribution range of the study area, we used the buffer analysis tool in ArcGIS to divide the study area into six levels. The closer an area is to a water system, the higher the possibility of a landslide is. Thus, an area ≤100 m away from a river was coded as 1. The greater the distance is, the lower the probability of a landslide is. Thus, an area >1000 m away from a river was coded as 6.

E.　Geology

Geological map data [50] provided by the Central Geological Survey of the Ministry of Economic Affairs were used to obtain geological names and rock properties corresponding to each grid. The strength grades of various geologies in the area were classified in accordance with the relationship between compressive strength and rock strength grades proposed by the International Society of Rock Mechanics [51]. A higher strength grade was coded as 1, whereas a lower strength grade was coded as 6. The strength grade was divided into six levels [1].

F.　Distance from the fault

Torkashvand et al. [52] reported that the areas within 500 m of a fault have a total landslide area of 39.18% and that the areas further than 3500 m from the fault have high slope stability. In addition, Guo et al. [53] indicated that when the distance from a fault exceeds 2500 m, it does not significantly affect the occurrence of landslides. Therefore, the present study used the buffer analysis tool in ArcGIS to divide the distance from the fault into three levels: less than 500 m, 500–3000 m, and more than 3000 m.

4.2.2. Disturbance Factor of Slope Land Utilization

Slope land use varies with time or space. Therefore, this study interpreted the aforementioned satellite images to determine the slope land use disturbance and obtain information on landslides in the study area for different periods. In accordance with the method reported by Chen et al. [1,43], we selected five items, namely road density, building density, crop density, green coverage rate, and bare land density, as slope land use disturbance

factors affecting the occurrence of landslides in the study area. We used the image classification module (segmentation and classification tools) in ArcGIS to calculate the proportions of slope land use disturbance factors in each basic grid (40 m × 40 m) of the study area after image interpretation. The size of the interpretation grid was equal to the resolution of each satellite image.

The method used to estimate the degree of slope land use disturbance under specific natural environmental conditions in the study area was based on the method reported by Chen et al. [43] and is presented in Equation (3).

$$I_{DLD} = {G_{DC}} \Big/ {G_{EC}} = {\sum (W_{DC} \cdot R_{DC})} \Big/ {\sum (W_{EC} \cdot R_{EC})'} \tag{3}$$

where the grade of the disturbance condition ($G_{DC}$) is the score of the aforementioned slope land use disturbance in each grid and the grade of the environmental conditions ($G_{EC}$) is the score of the natural environment of the sloping land in each grid. $G_{DC}$ is the ratio of the area of each slope land use disturbance factor in the basic grid, and $R_{EC}$ represents the coding of the natural environmental factors of each slope land in the basic grid.

### 4.2.3. Rain Trigger Factors

Rainfall data before and after a typhoon or rainstorm occurred in the study area in the past few years were obtained from the rain gauge station [46] of the Central Weather Bureau. The inverse distance weighted (IDW) method of the ArcGIS spatial interpolation calculation tool was used to estimate the distribution of the rainfall trigger indicators in the study area as a whole and in the basic grid.

### 4.2.4. Earthquake Trigger Factors

The earthquake factor index of landslides that occurred in the study area was determined as the PGA from the free-field strong seismic observation network of the Seismological Center of the Central Weather Bureau [54]. We used the IDW function of ArcGIS Spatial Analyst to estimate the distribution of seismic factors in the study area, and the results were evaluated for the basic grid.

### 4.3. Weight Analysis of Hazard-Causing Factors

This study used the RF method to analyze the explanatory power of the hazard factors and used the mean decrease accuracy of the RF analytical results to investigate the effects of the characteristic variables and calculate the explanatory power of each hazard factor. To determine the relationship between the slope land use disturbance factors and natural environmental factors, we used the aforementioned set of natural environmental factor classification codes. We used the aforementioned set of indicators to investigate the effect of the correlation between slope land use disturbance and natural environmental factors on landslides [55]. Building density, road density, and bare land density were all positively correlated with landslides, whereas crop density and green coverage were negatively correlated with landslides. Elevation, slope, aspect, and distance coding from a water system were negatively correlated with the occurrence of landslides, whereas distance coding from the fault and geological coding were positively correlated with the occurrence of landslides.

The average reduction in the precision of slope land use disturbance factors and natural environmental factors was divided by the total average reduction in precision to estimate the explanatory power of each hazard factor. After obtaining the square root and adding the correlation coefficient, we obtained positive and negative correlation values. The effects of the characteristic variables of slope land use disturbance factors and natural environmental factors are summarized in Table 2 (a) and Table 2 (b), respectively.

**Table 2.** Effects of the characteristic variables of slope land use disturbance factors and natural environmental factors.

| (a) Slope Land Use Disturbance Factor | | | |
|---|---|---|---|
| **Item** | **Degree of Influence of Characteristic Variables** | | |
| | **Mean Decrease Accuracy** | **Explanatory Power** | **Correlation Value** |
| Road Density | 39.88 | 0.15 | 0.39 |
| Building Density | 37.59 | 0.14 | 0.37 |
| Bare Density | 48.00 | 0.18 | 0.42 |
| Crop Density | 61.42 | 0.22 | −0.48 |
| Green Coverage | 84.35 | 0.31 | −0.56 |
| Total | 271.24 | 1 | |
| (b) Natural Environment Factors | | | |
| **Item** | Degree of Influence of Characteristic Variables | | |
| | Mean Decrease Accuracy | Explanatory Power | Correlation Value |
| Elevation Code | 103.09 | 0.13 | −0.36 |
| Slope Code | 235.93 | 0.29 | −0.54 |
| Aspect Code | 90.90 | 0.11 | −0.33 |
| Geology Code | 117.76 | 0.14 | 0.37 |
| Distance Code from Water system | 150.72 | 0.18 | −0.42 |
| Distance Code from Fault | 120.81 | 0.15 | 0.39 |
| Total | 819.21 | 1 | |

As shown in Table 2 (a), the larger the correlation value was, the stronger the effect of the factor on the occurrence of landslides was. The impact score of bare land density on landslide occurrence was 5. By contrast, the impact score of the green coverage rate with the smallest correlation value was 1. The impact scores of the slope land use disturbance factors on landslide occurrence are listed in Table 3.

**Table 3.** Impact scores of slope land use disturbance factors on landslides.

| Slope Use Disturbance Factor | Green Coverage | Crop Density | Building Density | Road Density | Bare Density |
|---|---|---|---|---|---|
| Score | 1 | 2 | 3 | 4 | 5 |

The aforementioned findings related to the hierarchical coding of natural environmental factors indicated that the higher the probability of a landslide was, the smaller the coding value was. According to the positive and negative correlation values of the natural environmental factors listed in Table 2 (b), the correlation value of the slope coding was the smallest. Therefore, the score of the impact of the slope coding on landslide occurrence was set to 6. By contrast, the correlation value of the distance from the fault coding was the highest, and the score of its impact on landslide occurrence was set to 1. The impact scores of the related natural factors on landslide occurrence are listed in Table 4.

**Table 4.** Impact scores of natural environmental factors on landslide occurrence.

| Natural Environment Factors | Distance Code from Fault | Geology Code | Aspect Code | Elevation Code | Distance Code from Water System | Slope Code |
|---|---|---|---|---|---|---|
| Score | 1 | 2 | 3 | 4 | 5 | 6 |

In this study, the $EAR \times I_{3Rmax}$-normalized data of each single rainfall occurrence and the PGA-normalized data of each single earthquake were integrated with the $EAR \times I_{3Rmax}$- and PGA-normalized data of postrainfall earthquakes and postearthquake rainfall. Then, an RF was used to analyze the degree of interpretation. Input data included the normalized values of $EAR \times I_{3Rmax}$ and the PGA of each basic grid under each rainfall occurrence or each earthquake in the study area and the landslide condition of the corresponding basic grid after each rainfall occurrence or each earthquake. The mean reduction in precision was used to determine the explanatory degree of characteristic variables and estimate the explanatory power of different trigger factors. The degree of influence of $EAR \times I_{3Rmax}$ and PGA characteristic variables under different trigger factors is summarized in Table 5. The explanatory power of each trigger factor item was set as the score value, and the postrainfall earthquake or postearthquake rainfall score value was obtained by adding up their individual explanatory powers. The effect of rainfall trigger factors on landslide occurrences in the study area was slightly higher than that of earthquake trigger factors.

**Table 5.** Degree of influence of $EAR \times I_{3Rmax}$ and PGA under different trigger factors.

| Trigger Factor | Index | Influence Degree of Characteristic Variables | | |
| --- | --- | --- | --- | --- |
| | | Mean Decrease Accuracy | Explanatory Power | Score |
| Single Rain | $EAR \times I_{3Rmax}$ | 90.97 | 0.27 | 0.27 |
| Single Earthquake | $PGA$ | 54.28 | 0.16 | 0.16 |
| Post-Earthquake Rainfall | $EAR \times I_{3Rmax}$ | 48.34 | 0.14 | 0.32 |
| | $PGA$ | 60.88 | 0.18 | |
| post-rainfall Earthquake | $EAR \times I_{3Rmax}$ | 45.50 | 0.13 | 0.25 |
| | $PGA$ | 39.34 | 0.12 | |
| | Total | 339.31 | 1 | |

*4.4. Establishment and Discussion of Hazard Indicators of the Interactive Correlation between Rainfall- and Earthquake-Induced Landslides*

4.4.1. Establishment of Hazard Indicators of Interaction between Rainfall and Earthquakes

This study explored the effect of the interaction between rainfall and earthquake trigger factors on landslide occurrences in slope land in the study area. This study referred to the degree of land development and utilization proposed by Chen et al. [1] and Chen et al. [56] and adjusted the hazard potential index to enable consideration of slope land disturbance. The $I_{HERI}$ of a basic grid number *i* after each rainfall occurrence or earthquake in the study area is defined as follows:

$$[I_{HERI}]_i = [(C_E + C_{RE}) \cdot F(T_E) + (C_R + C_{ER}) \cdot F(T_R)]_i \tag{4}$$

$F(T_R)$: Standardized values of rainfall factors in each basic grid
$F(T_E)$: Standardized values of seismic factors in each basic grid
$C_R$: Estimated score value for a single rainfall-induced landslide
$C_E$: Estimated score value for a single earthquake-induced landslide
$C_{RE}$: Score value calculated for a postrainfall earthquake-induced landslide
$C_{ER}$: Score value calculated for a postearthquake rainfall-induced landslide

where $F(T_R)$ or $F(T_E)$ can be obtained from the spatial distribution data of the trigger factors ($EAR \times I_{3Rmax}$ or PGA) of each rainfall occurrence or each earthquake in the study area. The values of $C_R$, $C_E$, $C_{RE}$, and $C_{ER}$ can be calculated separately using the aforementioned RF algorithm through a weight analysis. The trigger factors of single rainfall, single earthquake, earthquake after a previous rainfall event, and rainfall after a previous earthquake event induced landslides in the slope land in the study area. The larger the $I_{HERI}$ was, the higher the possibility of the interaction between rainfall and earthquakes inducing a landslide was.

To investigate the relationship between the grid ratio of the landslides (the number of basic grids with landslides divided by the number of basic grids without landslides) induced by the aforementioned rainfall or earthquake events (13 in total) in the study area and $I_{HERI}$, we first used SPSS Cluster Analysis [55] to categorize the $I_{HERI}$ values into five levels and then plotted the graph for the $I_{HERI}$ against the landslide grid ratio (Figure 2). The landslide grid ratio in the study area tended to increase with an increase in the degree of interaction between rainfall and earthquake events.

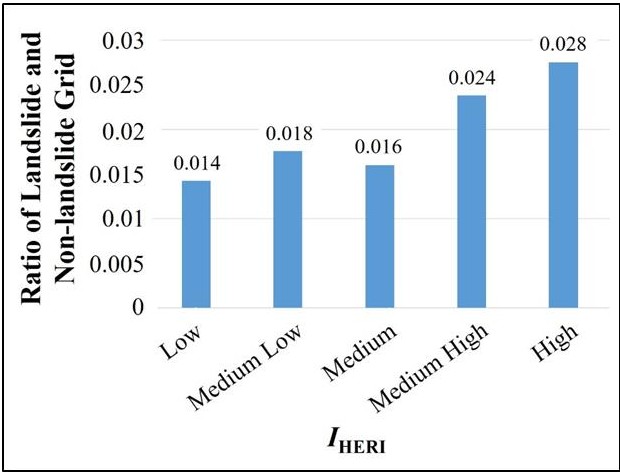

**Figure 2.** The hazard index $I_{HERI}$ against landslide grid ratio.

### 4.4.2. Classification of Slope Land Use Disturbance Degree and Its Influence on Landslides in the Study Area

The data collected after 13 rainfall- or earthquake-induced landslides in the study area were merged, and data for 2,339,322 basic grids were obtained. Equation (3) was used to estimate the $I_{DLD}$ of each basic grid. By using SPSS Cluster Analysis, we categorized the $I_{DLD}$ values into five grades. We plotted the $I_{DLD}$ values against the 13 rainfall- or earthquake-induced landslide grid ratios in the study area (the number of basic grids with landslides divided by the number of basic grids where landslides occur) to determine the degree of slope land use disturbance (Figure 3). The landslide grid ratio of the slope land with a low degree of disturbance was 0.006, whereas that of the slope land with a high degree of disturbance was as high as 0.93. The results indicated that the landslide grid ratio of the slope land in the study area increased with the degree of slope land use disturbance.

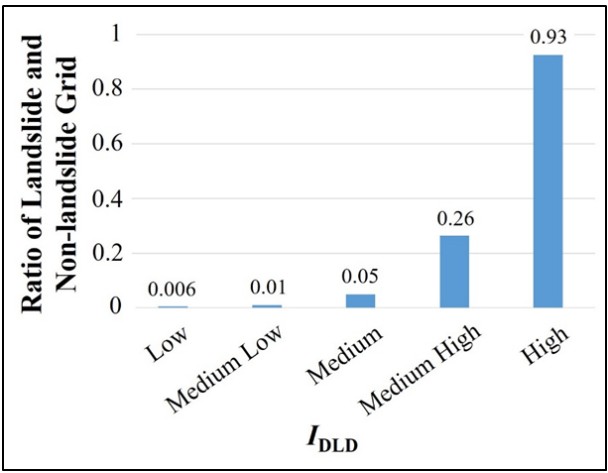

**Figure 3.** The degree of land disturbance $I_{DLD}$ against landslide grid ratio.

### 4.4.3. Interaction between the Hazard Indexes of Rainfall- and Earthquake-Induced Landslides and the Index of the Degree of Slope Land Use Disturbance

This study investigated the relationships among $I_{HERI}$, $I_{DLD}$, and landslide occurrence in each basic grid in the study area. A schematic of the interval settings is presented in Figure 4 and the schematic obtained after dividing the study area into 9 equidistant interval grids (from letter A to I) is presented in Figure 5. The red ○'s and gray ✕'s in the figure represent basic grids with and without landslides, respectively. The basic grid with landslides represents the total number of landslides induced by all 13 rainfall or earthquake events in the study area, which yielded a total of 37,197 records. The grid points in the figure where a landslide did not occur were determined using a 1:1 ratio, which was obtained through random sampling from all basic grids where a landslide did not occur.

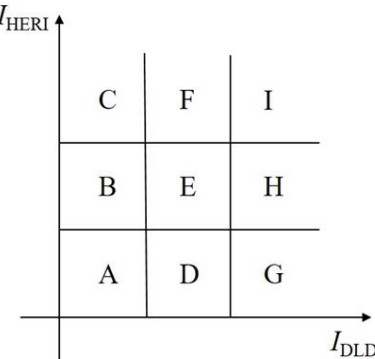

**Figure 4.** Schematic of the interval setting method of the relationships among the $I_{HERI}$, $I_{DLD}$, and landslide occurrence of each basic grid.

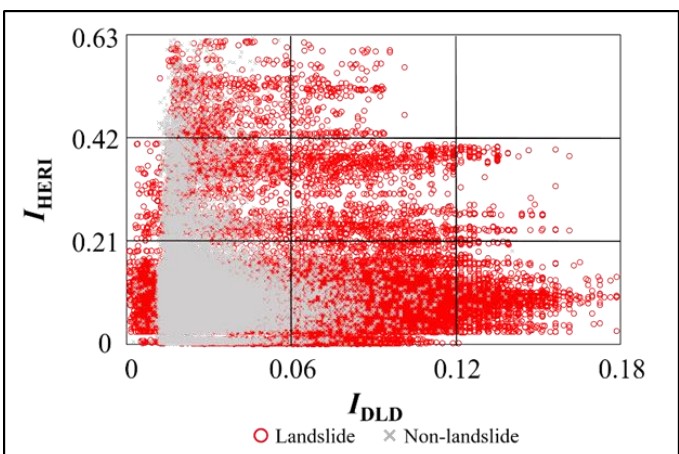

**Figure 5.** Relationships among the $I_{HERI}$, $I_{DLD}$, and landslide occurrence after various rainfall and earthquake events in the study area.

The number of basic grids and the ratio of collapsed grids to noncollapsed grids for each interval are presented in Figure 5 and summarized in Table 6. In each interval, with the exception of I, which did not have data, only the A and B grids without landslides were larger than the grids with landslides. The ratio of landslide grids (the ratio of the number of landslide grids to the number of nonlandslide grids or the ratio of the number of landslide grids to the total number of grids in an interval) sequentially increased from A to H. When the $I_{HERI}$ or $I_{DLD}$ was larger, the induced landslide ratio increased. When the degree of slope land use disturbance was similar in the study area, the landslide ratio, which indicated the probability of a landslide, increased with the $I_{HERI}$ value. Similarly, when the $I_{HERI}$ was similar in the study area, the probability of a landslide increased with the degree of slope land use disturbance. The interaction between rainfall and earthquake

events in the study area indicates that the effect of the degree of slope land use disturbance on landslide occurrence is notable.

**Table 6.** The $I_{HERI}$, $I_{DLD}$, and number of grids with and without landslides and the ratio of the landslide grids in each interval corresponding to the occurrence of landslides.

| Interval No. | Number of Grids with Landslide | Number of Grids without Landslide | Landslide Grid Ratio | |
|---|---|---|---|---|
| | | | Number of Grids with Landslide/ Number of Grids without Landslide | Number of Grids with Landslide/ Total Number of Grids in the Interval |
| A | 16,373 | 33,326 | 0.49 | 0.33 |
| B | 1656 | 2297 | 0.72 | 0.42 |
| C | 644 | 489 | 1.32 | 0.57 |
| D | 14,795 | 1001 | 14.80 | 0.94 |
| E | 1132 | 47 | 24.10 | 0.96 |
| F | 216 | 6 | 36.00 | 0.97 |
| G | 2205 | 30 | 73.50 | 0.99 |
| H | 176 | 1 | 176.00 | 0.99 |
| I | 0 | 0 | — | — |

*4.5. Establishment and Verification of Rainfall- and Earthquake-Induced Landslide Potential Assessment Models and Potential Map Drawing*

4.5.1. Establishment of a Landslide Potential Assessment Model

On the basis of the data collected on land surface changes and disasters after 13 rainfall or earthquake events in the study area between 2004 and 2018 and with reference to Wang [39], we used the RF algorithm developed in R language to establish a landslide potential assessment model with consideration of four trigger factors. After the collection and analysis of 839,047 records of single rainfall events (Typhoon Mindulle, 0726 heavy rain, Nanmadu typhoon, 0520 heavy rain, and Megi typhoon), 1,033,319 records of single earthquake events (0226 earthquake, 0602 earthquake, 0206 earthquake, 1122 earthquake, and 0320 earthquake), 163,189 records of postrainfall earthquake events (1105 earthquake after Morakot and 1108 earthquake after Fanapi), and 303,767 records of postearthquake rainfall events (Kalmaegi after 0305 earthquake), with reference to Chen et al. [56], we randomly sampled the records without landslides on the basis of the number of landslides and then constructed training and testing data sets at a ratio of 7:3. By performing an error matrix analysis, we determined the PA and OA and used them as evaluation factors for establishing precision criteria. The results are summarized in Table 7.

**Table 7.** Producer accuracy and overall accuracy for training and testing data sets.

| Trigger Factor / Accuracy | (a) Single Rainfall | | (b) Single Earthquake | |
|---|---|---|---|---|
| | Training | Testing | Training | Testing |
| *PA* of Grids with Landslide | 91.61% | 83.14% | 98.66% | 95.96% |
| *PA* of Grids without Landslide | 99.40% | 92.17% | 99.90% | 95.63% |
| Overall Accuracy | 100% | 89.84% | 99.92% | 95.70% |
| Trigger Factor / Accuracy | (c) Post-rainfall Earthquake | | (d) Post-earthquake Rainfall | |
| | Training | Testing | Training | Testing |
| *PA* of Grids with Landslide | 98.23% | 87.02% | 96.26% | 89.99% |
| *PA* of Grids without Landslide | 99.68% | 88.01% | 98.21% | 92.22% |
| Overall Accuracy | 100% | 88.27% | 99.84% | 91.67% |

The PA of both the training and testing sets was above 83%, and the average OA was 95.7. The finding indicated that the evaluation model could achieve a high degree of accuracy for landslide classification. In addition, both the PA and OA of potential landslides induced by a single rainfall event, a single earthquake, a postrainfall earthquake, and postearthquake rainfall were above 83%, indicating excellent accuracy (Tables 8–11).

**Table 8.** Evaluation accuracy of single rainfall-induced landslide potential.

| Single Typhoon or Rainfall | | Typhoon Mindulle | | 0726 Heavy Rain | | Typhoon Nanmadol | |
|---|---|---|---|---|---|---|---|
| Coverage | | Alishan Township | | Dapu Township | | Dapu Township | |
| | | Evaluation Result | | | | | |
| | | Landslide | Non-landslide | Landslide | Non-landslide | Landslide | Non-landslide |
| Actual Situation | Landslide | 6083 | 919 | 697 | 2 | 522 | 1 |
| | Non-landslide | 20,948 | 225,031 | 3993 | 77,505 | 2279 | 76,647 |
| *PA* of Grids with Landslide | | 86.88% | | 99.71% | | 99.81% | |
| *PA* of Grids without Landslide | | 91.48% | | 95.10% | | 97.11% | |
| Overall Accuracy | | 91.36% | | 95.14% | | 97.13% | |
| Single Rainfall or Earthquake | | 0520 Heavy Rain | | 0520 Heavy Rain | | Typhoon Megi | |
| Coverage | | Alishan Township | | Dapu Township | | Dapu Township | |
| | | Evaluation Result | | | | | |
| | | Landslide | Non-landslide | Landslide | Non-landslide | Landslide | Non-landslide |
| Actual | Landslide | 3958 | 467 | 1012 | 177 | 577 | 38 |
| | Non-landslide | 18,868 | 242,051 | 5341 | 70,094 | 3296 | 78,541 |
| *PA* of Grids with Landslide | | 89.45% | | 85.11% | | 93.82% | |
| *PA* of Grids without Landslide | | 92.77% | | 92.92% | | 95.96% | |
| Overall Accuracy | | 92.71% | | 92.80% | | 95.96% | |

**Table 9.** Evaluation accuracy of single earthquake-induced landslide potential.

| Single Earthquake | | 0226 Earthquake | | 0602 Earthquake | | 0206 Earthquake | |
|---|---|---|---|---|---|---|---|
| Coverage | | Dapu Township | | Alishan Township | | Alishan Township | |
| | | Evaluation Result | | | | | |
| | | Landslide | Non-landslide | Landslide | Non-landslide | Landslide | Non-landslide |
| Actual Situation | Landslide | 687 | 22 | 6602 | 157 | 3839 | 122 |
| | Non-landslide | 3155 | 76,990 | 10,587 | 236,645 | 11,263 | 249,499 |
| *PA* of Grids with Landslide | | 96.90% | | 97.68% | | 96.92% | |
| *PA* of Grids without Landslide | | 96.06% | | 95.72% | | 95.68% | |
| Overall Accuracy | | 96.07% | | 95.77% | | 95.70% | |
| Single Earthquake | | 0206 Earthquake | | 1122 Earthquake | | 0320 Earthquake | |
| Coverage | | Dapu Township | | Alishan Township | | Dapu Township | |
| | | Evaluation Result | | | | | |
| | | Landslide | Non-landslide | Landslide | Non-landslide | Landslide | Non-landslide |
| Actual Situation | Landslide | 475 | 27 | 3139 | 6 | 484 | 5 |
| | Non-landslide | 2197 | 79,731 | 14,714 | 253,350 | 3219 | 76,404 |
| *PA* of Grids with Landslide | | 94.62% | | 99.81% | | 98.98% | |
| *PA* of Grids without Landslide | | 97.32% | | 94.51% | | 95.96% | |
| Overall Accuracy | | 97.30% | | 94.57% | | 95.98% | |

**Table 10.** Evaluation accuracy of postrainfall earthquake-induced landslide potential.

| Post-Rainfall Earthquake | | 1105 Earthquake after Typhoon Morakot | | 1108 Earthquake after Typhoon Fanapi | |
|---|---|---|---|---|---|
| Coverage | | Dapu Township | | Dapu Township | |
| | | Evaluation Result | | | |
| | | Landslide | Non-landslide | Landslide | Non-landslide |
| Actual Situation | Landslide | 1568 | 116 | 962 | 24 |
| | Non-landslide | 12,975 | 67,610 | 5102 | 74,832 |
| *PA* of Grids with Landslide | | 93.11% | | 97.57% | |
| *PA* of Grids without Landslide | | 83.90% | | 93.62% | |
| Overall Accuracy | | 84.09% | | 93.67% | |

**Table 11.** Evaluation accuracy of postearthquake rainfall-induced landslide potential.

| Post-Earthquake Rainfall | | Typhoon Kalmaegi after 0305 Earthquake | | Typhoon Kalmaegi after 0305 Earthquake | |
|---|---|---|---|---|---|
| Coverage | | Alishan Township | | Dapu Township | |
| | | Evaluation Result | | | |
| | | Landslide | Non-landslide | Landslide | Non-landslide |
| Actual Situation | Landslide | 3348 | 184 | 910 | 67 |
| | Non-landslide | 15,377 | 221,588 | 10,275 | 52,018 |
| *PA* of Grids with Landslide | | 94.79% | | 93.14% | |
| *PA* of Grids without Landslide | | 93.51% | | 83.51% | |
| Overall Accuracy | | 93.53% | | 83.65% | |

### 4.5.2. Verification of the Landslide Potential Assessment Model and Drawing of a Potential Map

We used the ROC to determine the effectiveness of applying the established landslide potential assessment model in the study area. We employed the AUC as a reference standard. The AUC values of the results for landslides induced by a single rainfall event, a single earthquake, a postrainfall earthquake, and postearthquake rainfall in the study area were 0.95, 0.99, 0.94, and 0.97, respectively. The prediction for induced landslides was highly accurate. We used the equal interval classification method of ArcGIS to classify the values for potential landslides induced by rainfall or earthquakes into four categories: 0–0.25, 0.25–0.5, 0.5–0.75, and 0.75–1. These categories correspond to low potential, medium-low potential, medium-high potential, and high potential, respectively. The higher the potential value is, the greater the probability of a landslide occurrence is. Figures 6–9 present the potential maps of the study area after the occurrence of landslides induced by a single rainfall event, a single earthquake, a postrainfall earthquake, and postearthquake rainfall.



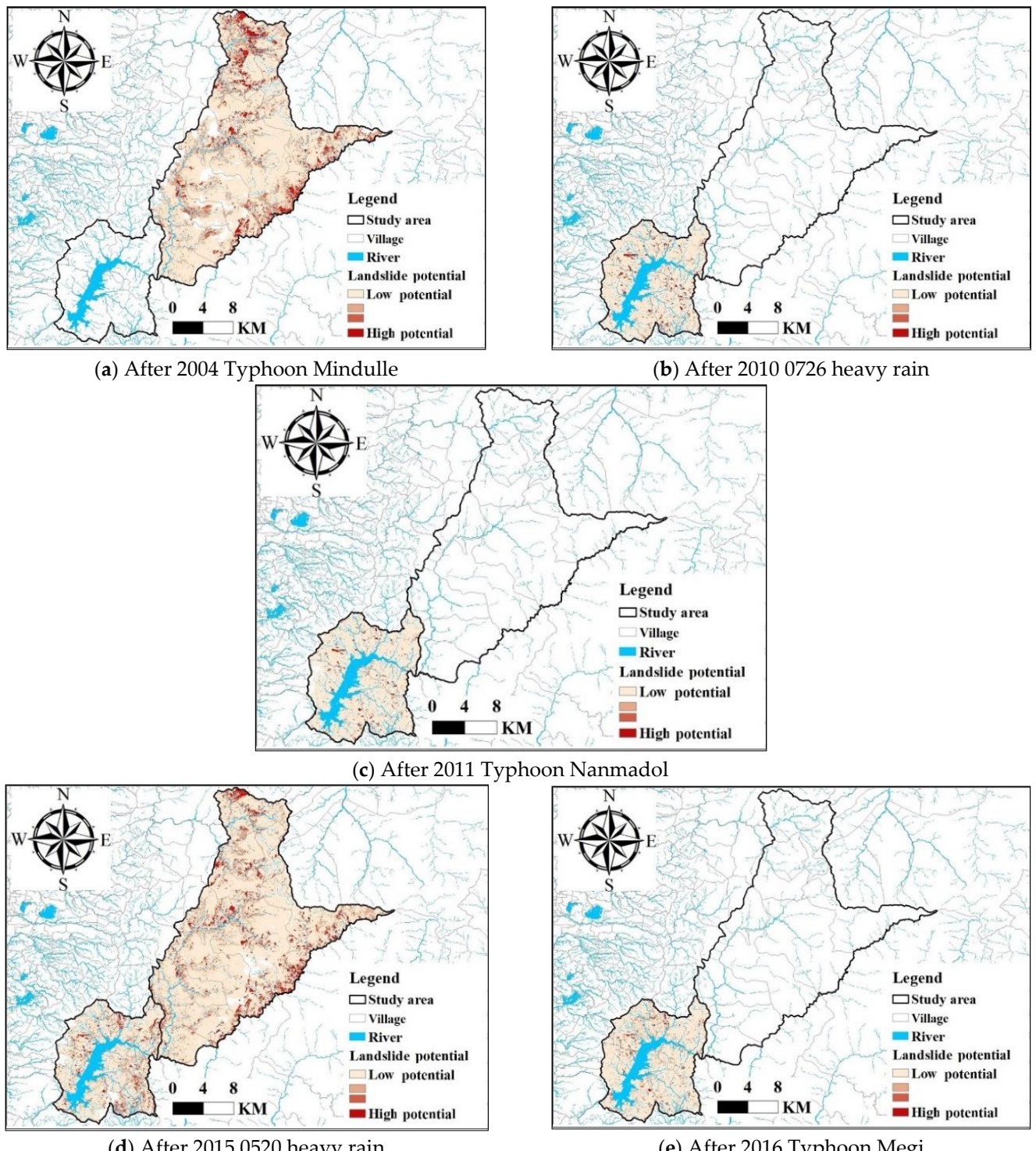

(**a**) After 2004 Typhoon Mindulle                    (**b**) After 2010 0726 heavy rain

(**c**) After 2011 Typhoon Nanmadol

(**d**) After 2015 0520 heavy rain                    (**e**) After 2016 Typhoon Megi

**Figure 6.** Single rainfall event-induced landslide potential.

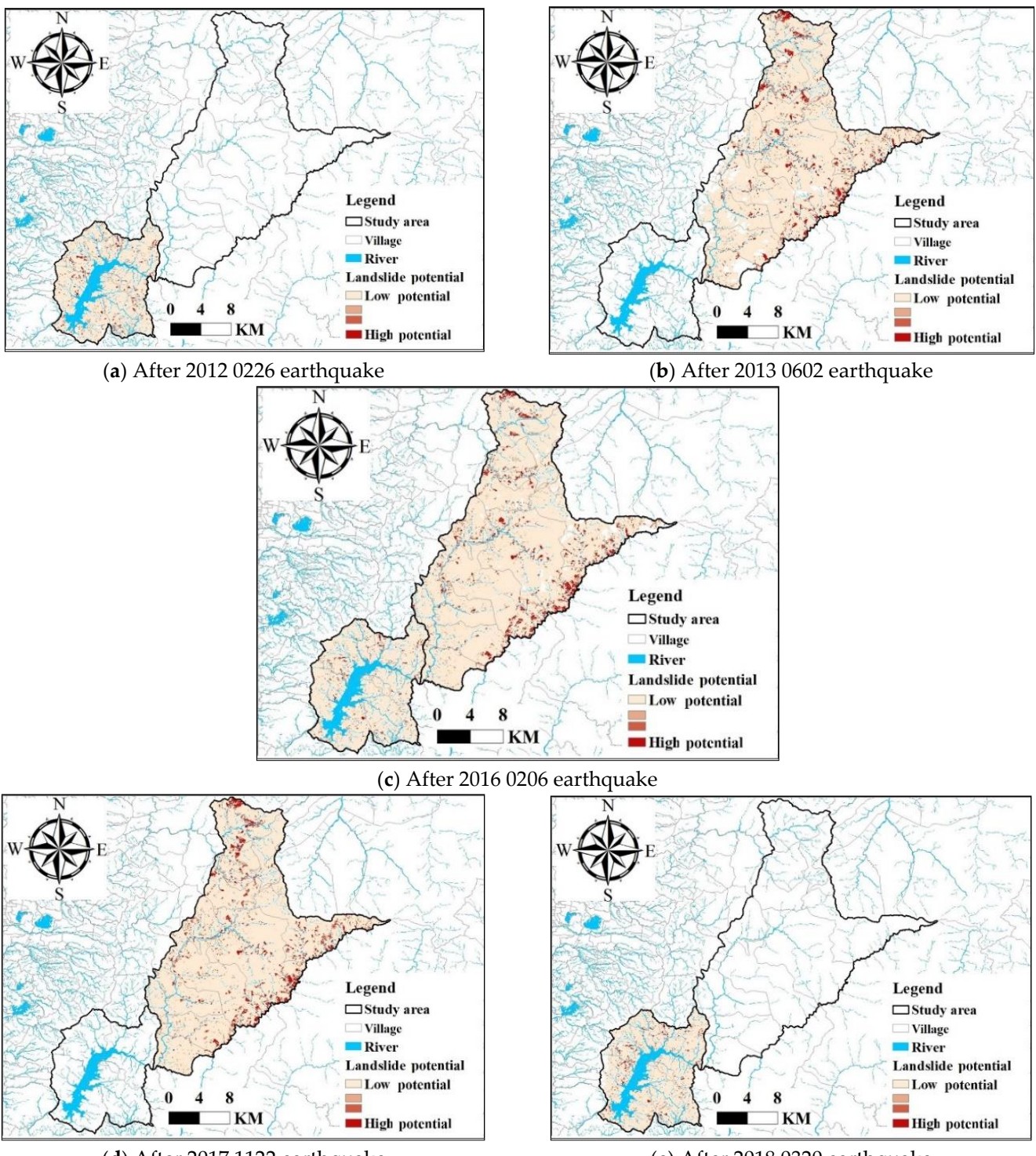

(**a**) After 2012 0226 earthquake        (**b**) After 2013 0602 earthquake

(**c**) After 2016 0206 earthquake

(**d**) After 2017 1122 earthquake        (**e**) After 2018 0320 earthquake

**Figure 7.** Single earthquake-induced landslide potential.

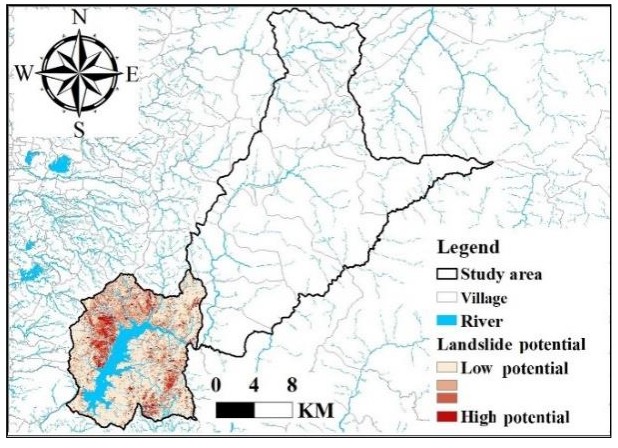

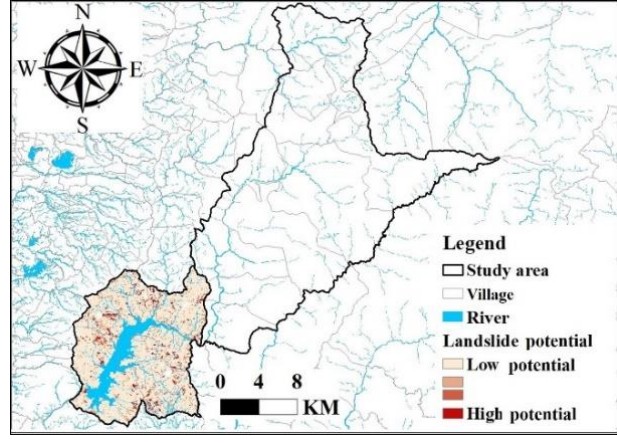

(**a**) 1105 earthquake after 2009 Typhoon Morakot

(**b**) 1108 earthquake after 2010 Typhoon Fanapi

**Figure 8.** Postrainfall earthquake-induced landslide potential.

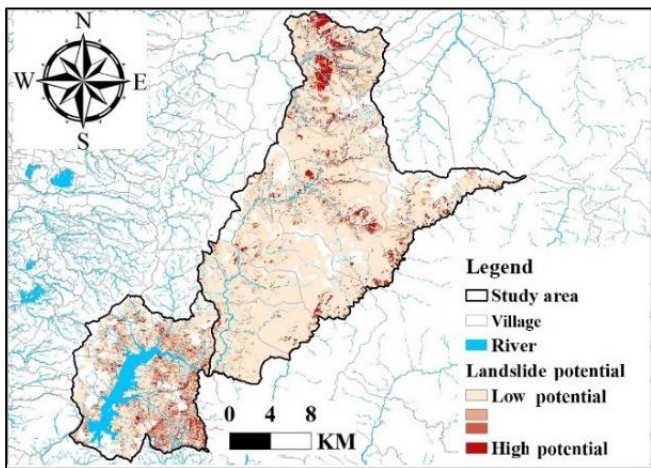

Typhoon Kalmaegi after 2008 0305 earthquake

**Figure 9.** Postearthquake rainfall-induced landslide potential.

This study referred to information provided by the Central Geological Survey [50] of the Ministry of Economic Affairs and the Soil and Water Conservation Bureau [57] of the Executive Yuan to obtain a distribution map of the historical disaster locations in the study area and an overlay map of potential landslides induced by the aforementioned rainfall and earthquake events. The landslide potential maps of Alishan Township and Dapu Township in the study area overlapped with historical landslide range results, as illustrated in Figures 10 and 11, respectively.

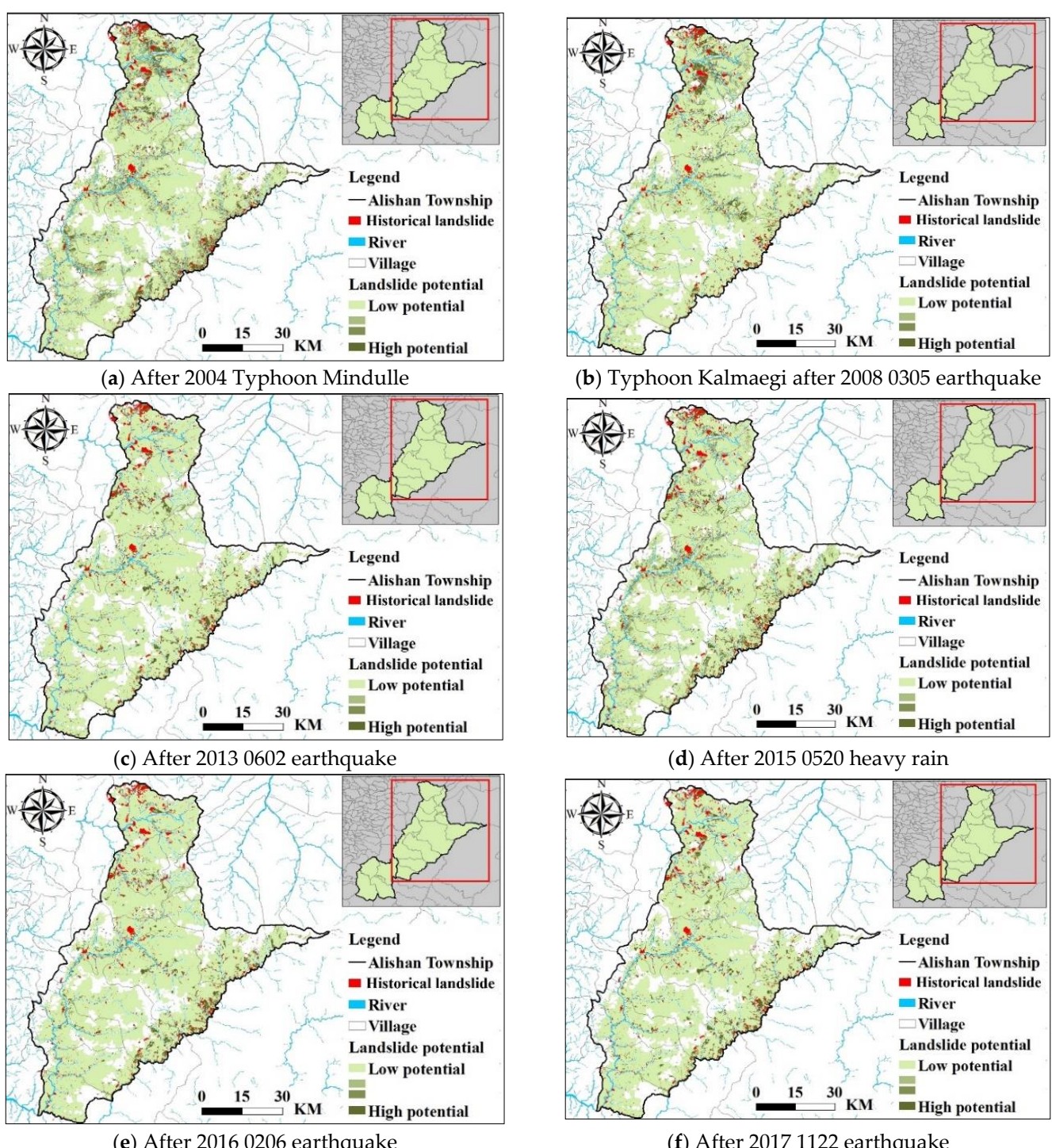

(**a**) After 2004 Typhoon Mindulle

(**b**) Typhoon Kalmaegi after 2008 0305 earthquake

(**c**) After 2013 0602 earthquake

(**d**) After 2015 0520 heavy rain

(**e**) After 2016 0206 earthquake

(**f**) After 2017 1122 earthquake

**Figure 10.** Landslide potential map of Alishan Township after rainfall and earthquakes overlapped by historical landslides.

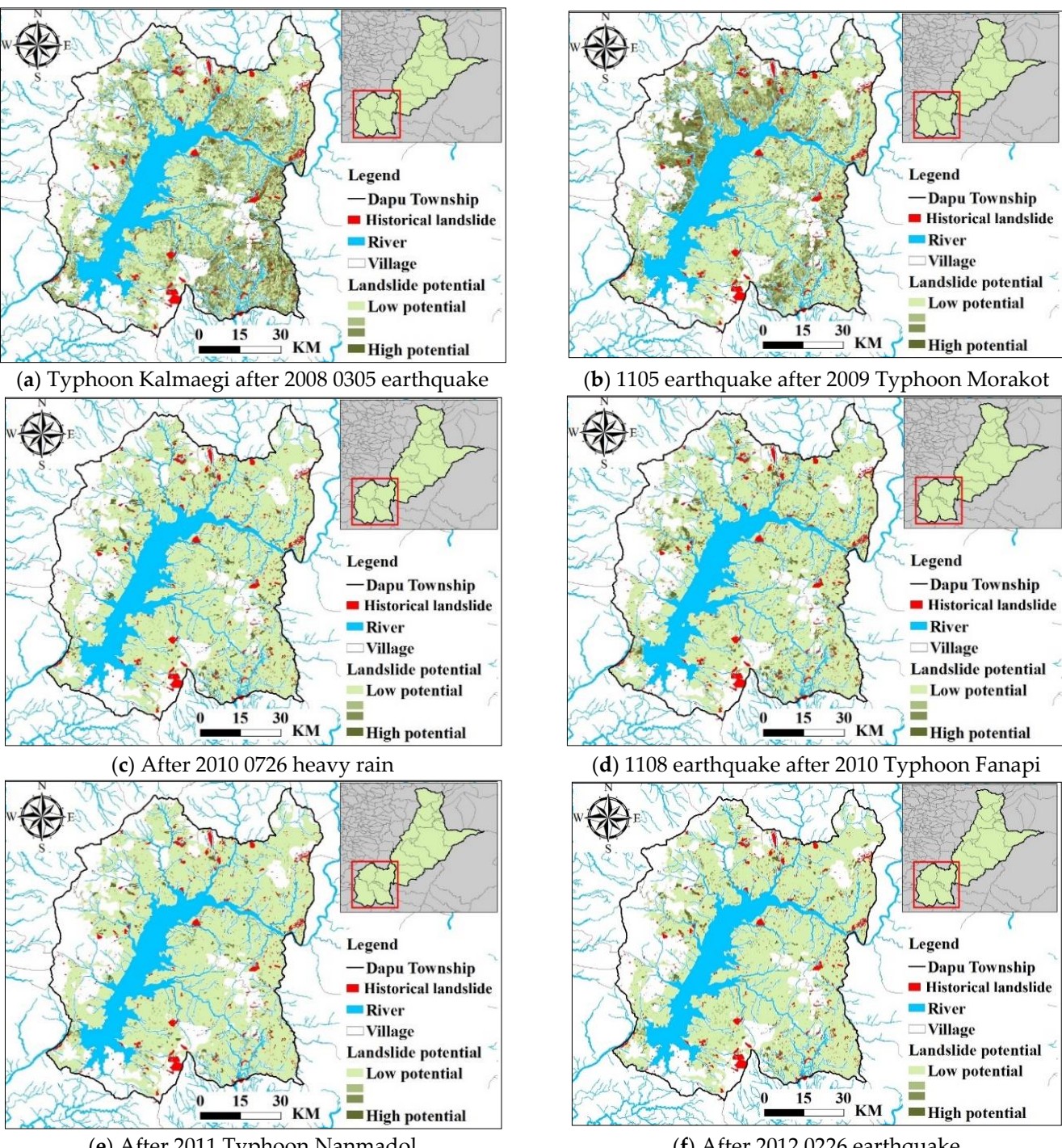

(**a**) Typhoon Kalmaegi after 2008 0305 earthquake

(**b**) 1105 earthquake after 2009 Typhoon Morakot

(**c**) After 2010 0726 heavy rain

(**d**) 1108 earthquake after 2010 Typhoon Fanapi

(**e**) After 2011 Typhoon Nanmadol

(**f**) After 2012 0226 earthquake

**Figure 11.** *Cont*.

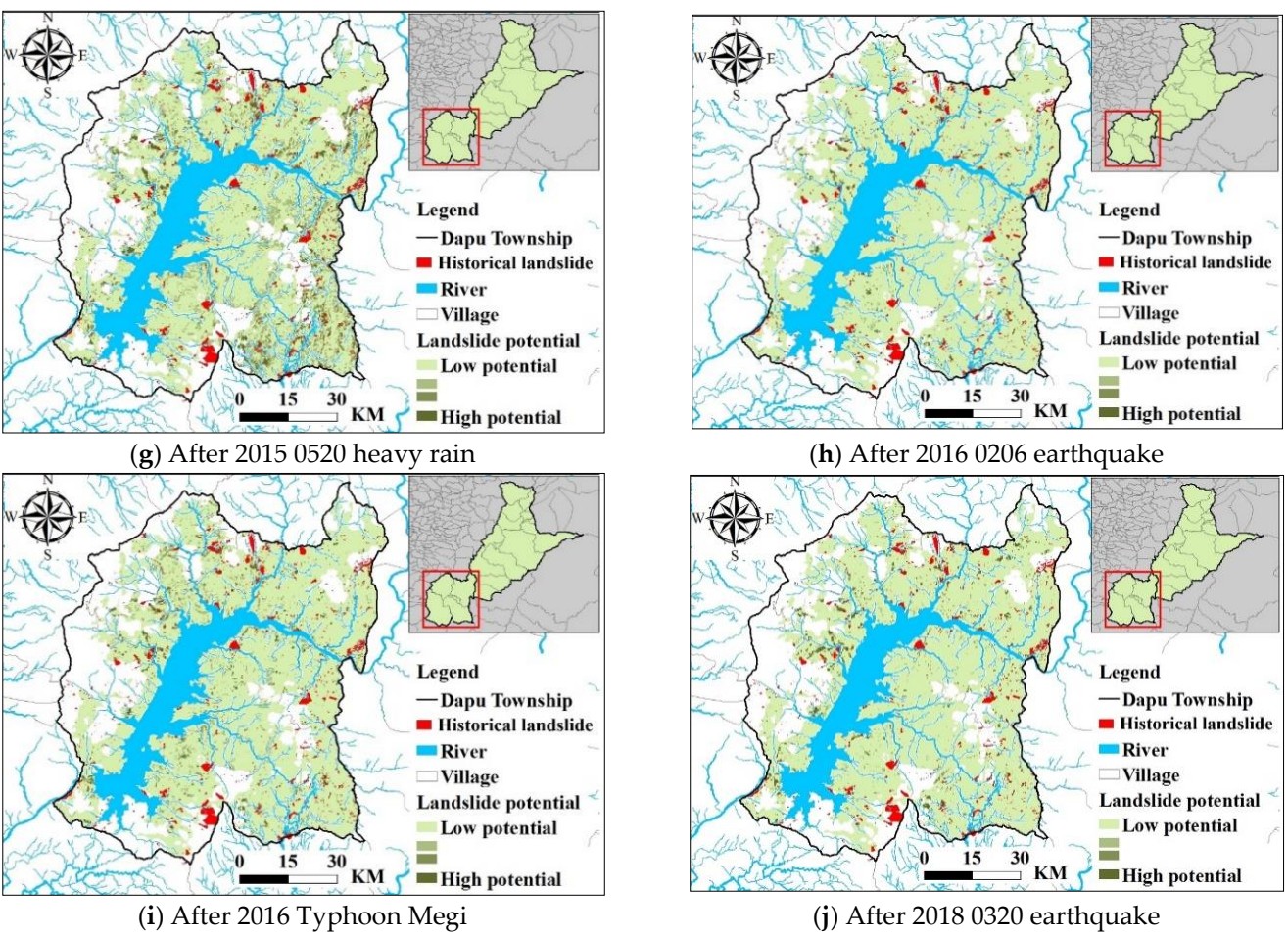

(**g**) After 2015 0520 heavy rain

(**h**) After 2016 0206 earthquake

(**i**) After 2016 Typhoon Megi

(**j**) After 2018 0320 earthquake

**Figure 11.** Landslide potential map of Dapu Township after rainfall or earthquakes overlapped by historical landslides.

In this study, we calculated the landslide area ratios of medium–high-level landslides induced by rainfall and earthquakes in Alishan Township and Dapu Township, respectively, to determine the degree of damage caused by different trigger factors. The area ratio of medium–high-level landslides was calculated as the area of historical disaster regions with landslide potential values of >0.5 divided by the total historical disaster area. To prevent cloud cover or shadows from affecting the results, we combined image grids affected by cloud cover or shadows before and after the 13 rainfall or earthquake events in the study area. Thus, the total grid number of satellite images before and after the rainfall and earthquake events was identical. The area ratios of medium–high-level landslides induced by rainfall and earthquakes in Alishan Township are shown in Table 12 and Figure 12, respectively, and those of Dapu Township are presented in Table 13 and Figure 13, respectively. With the exception of the area ratios of landsides induced by Typhoon Mindulle in 2004 in Alishan Township and 0520 heavy rainfall in Dapu Township, the landslide areas affected by the interaction between rainfall and earthquakes were large. A larger area ratio of medium–high-level landslides was observed in Alishan Township after typhoon Minduli in 2004 and Dapu Township after 0520 heavy rainfall. This may be because the average amount of rainfall was higher than that during trigger events (Table 14), which caused the probability of landslides during rainfall events to be higher than that during the dual trigger events. In addition, on the basis of the results presented in Tables 12–14, we plotted the trigger index values ($EAR \times I_{3Rmax}$ or PGA) against the landslide area ratios of medium–high-level landslides induced by a single rainfall or a single earthquake event in the study area (Figures 14 and 15, respectively). Irrespective of whether the average value of the $EAR \times I_{3Rmax}$ of a single rainfall event was larger or

the maximum value of the PGA of a single earthquake was larger, the landslide area ratio of the medium–high-level landslides exhibited an increasing trend. This finding indicates that the landslide potential estimated in this study is reasonable.

**Table 12.** Landslide area ratios of medium–high-level landslides induced by rainfall and earthquakes in Alishan Township.

| Trigger / Landslide Area or Area Ratio | Single Rainfall | | Single Earthquake | | | Post-Earthquake Rainfall |
|---|---|---|---|---|---|---|
| | **Typhoon Mindulle** | **0520 Heavy Rainfall** | **0602 Earthquake** | **0206 Earthquake** | **1122 Earthquake** | **Typhoon Kalmaegi after 0305 Earthquake** |
| Total area of historical disaster areas (hectares) | 1394.6 | 1394.6 | 1394.6 | 1394.6 | 1394.6 | 1394.6 |
| Area (hectares) in historical disaster area with medium-high landslide potential | 627.9 | 480.1 | 441.4 | 397.3 | 414.7 | 509.1 |
| Landslide area ratio of medium-high landslide potential | 0.45 | 0.344 | 0.317 | 0.285 | 0.297 | 0.365 |

**Table 13.** Landslide area ratios of medium–high-level landslides induced by rainfall and earthquakes in Dapu Township.

| Trigger / Landslide Area or Area Ratio | Single Rainfall | | | | Single Earthquake | | | Post-Earthquake Rainfall | Post-Rainfall Earthquake | |
|---|---|---|---|---|---|---|---|---|---|---|
| | **0726 Heavy Rain** | **Typhoon Nan-madol** | **0520 Heavy Rain** | **Typhoon Megi** | **0226 Earthquake** | **0206 Earthquake** | **0320 Earthquake** | **Typhoon Kalmaegi after 0305 Earthquake** | **1105 Earthquake after Typhoon Morakot** | **1108 Earthquake after Typhoon Fanapi** |
| Total area of historical disaster areas (hectares) | 482.6 | 482.6 | 482.6 | 482.6 | 482.6 | 482.6 | 482.6 | 482.6 | 482.6 | 482.6 |
| Area (hectares) in historical disaster area with medium-high landslide potential | 78 | 68.2 | 100.8 | 91.3 | 72.4 | 78.3 | 84.7 | 152.9 | 132.9 | 99.4 |
| Landslide area ratio of medium-high landslide potential | 0.162 | 0.141 | 0.209 | 0.189 | 0.15 | 0.162 | 0.176 | 0.317 | 0.275 | 0.206 |

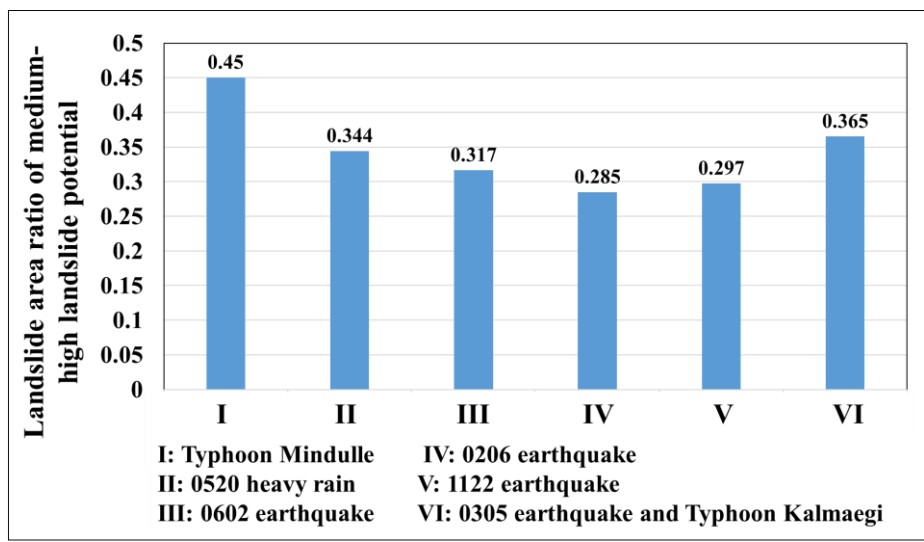

**Figure 12.** Landslide area ratio of medium–high-level landslides induced by rainfall and earthquakes in Alishan Township.

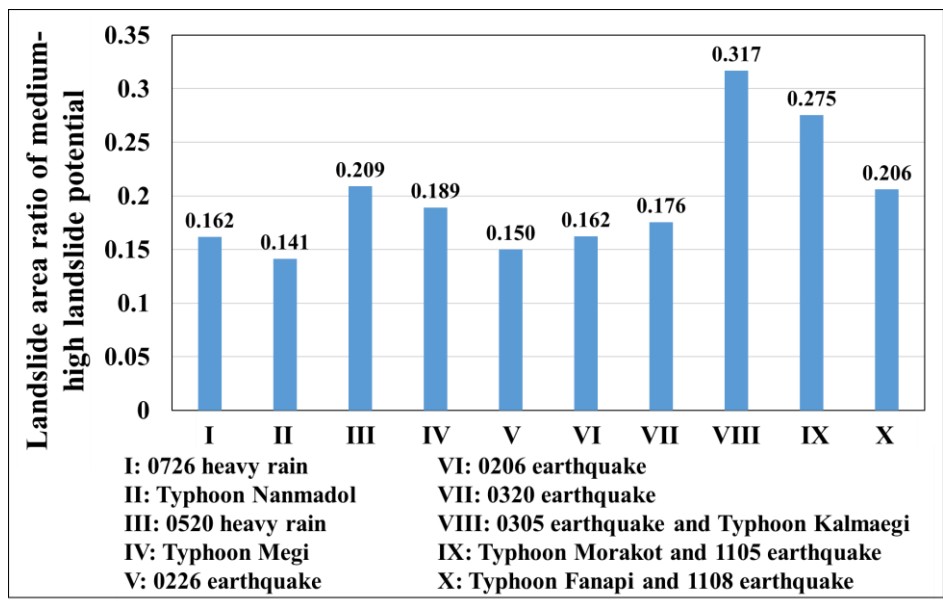

**Figure 13.** Landslide area ratio of medium–high-level landslides induced by rainfall and earthquakes in Dapu Township.

**Table 14.** $EAR \times I_{3Rmax}$ and PGA after rainfall and earthquakes.

| (a) Alishan Township | | | | | | |
|---|---|---|---|---|---|---|
| **Rainfall or (and) Earthquake** | **$EAR \times I_{3Rmax}$ (mm²/3-h)** | | | **PGA (gal)** | | |
| | **Maximum** | **Minimum** | **Mean** | **Maximum** | **Minimum** | **Mean** |
| Typhoon Mindulle | 138,449 | 39,993 | 86,407 | | | |
| 0520 Heavy Rainfall | 85,608 | 19,767 | 45,671 | | | |
| 0602 Earthquake | | | | 445 | 34 | 116 |
| 0206 Earthquake | | | | 212 | 52 | 126 |
| 1122 Earthquake | | | | 318 | 32 | 94 |
| Typhoon Kalmaegi after 0305 Earthquake | 131,572 | 18,518 | 82,081 | 88 | 7 | 35 |
| (b) Dapu Township | | | | | | |
| Rainfall or (and) Earthquake | **$EAR \times I_{3Rmax}$ (mm²/3-h)** | | | *PGA* (gal) | | |
| | Maximum | Minimum | Mean | Maximum | Minimum | Mean |
| 0726 Heavy Rainfall | 23,269 | 4713 | 11,161 | | | |
| Typhoon Nanmadol | 10,299 | 4856 | 7259 | | | |
| 0520 Heavy Rainfall | 67,375 | 40,667 | 58,007 | | | |
| Typhoon Megi | 78,593 | 20,659 | 39,822 | | | |
| 0226 Earthquake | | | | 68 | 25 | 55 |
| 0206 Earthquake | | | | 185 | 72 | 138 |
| 0320 Earthquake | | | | 243 | 77 | 146 |
| Typhoon Kalmaegi after 0305 Earthquake | 193,711 | 115,644 | 164,877 | 169 | 13 | 55 |
| 1105 Earthquake After Typhoon Morakot | 437,136 | 158,541 | 295,497 | 68 | 16 | 35 |
| 1108 Earthquake after Typhoon Fanapi | 35,863 | 9268 | 20,962 | 208 | 23 | 69 |

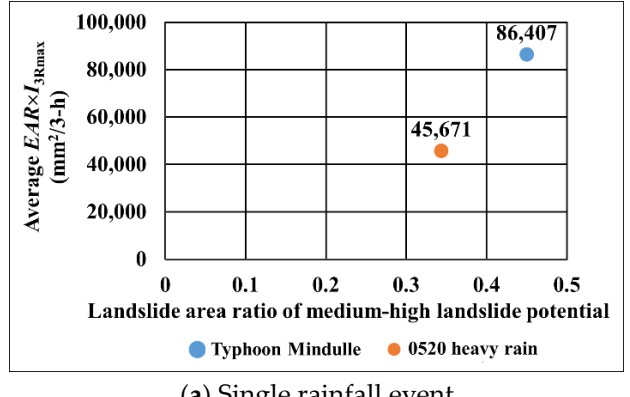

(**a**) Single rainfall event

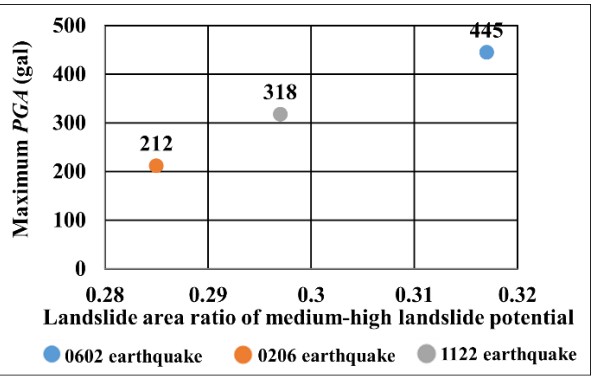

(**b**) Single earthquake event

**Figure 14.** Landslide area ratios of medium–high-level landslides against average $EAR \times I_{3Rmax}$ and maximum PGA induced by a single rainfall or a single earthquake event in Alishan Township.

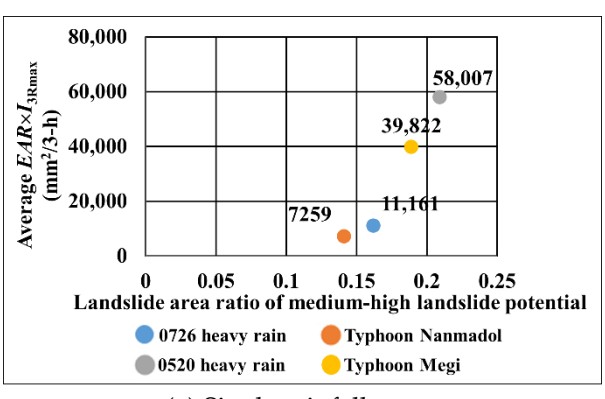

(**a**) Single rainfall event

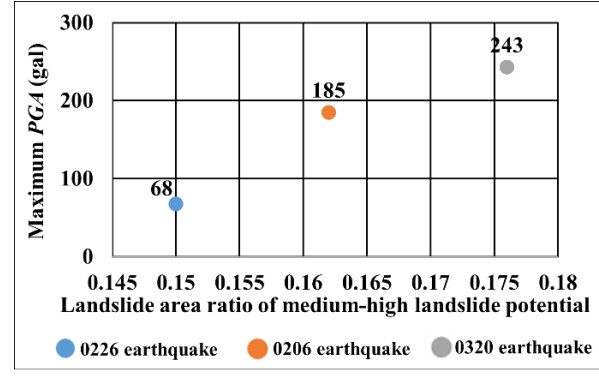

(**b**) Single earthquake event

**Figure 15.** Landslide area ratios of medium–high-level landslides against average $EAR \times I_{3Rmax}$ and maximum PGA induced by a single rainfall or a single earthquake event in Dapu Township.

## 5. Conclusions

This study used the random tree image classification module of a GIS platform to interpret the satellite images of the study area before and after the occurrence of landslides induced by four types of events, namely a single rainfall event, a single earthquake, a postrainfall earthquake, and postearthquake rainfall. We used the texture analysis module of ERDAS IMAGINE to perform image interpretation and obtain information on surface changes and landslide areas in the study area. The major mapped landslides are debris slides. The average Kappa value (0.66) and average OA (70.1%) of the 37 satellite images before and after the 13 rainfall and earthquake events in the study area, which were determined using seven interpretive classifications, namely water, roads, buildings, cash crops, green coverage, river channels, and bare land, reveal that the interpretation accuracy was medium to medium high. In addition, we established four landslide potential assessment models for landslides induced by different trigger factors by using an RF. We established a hazard index ($I_{HERI}$) for landslides induced by postrainfall earthquakes and postearthquake rainfall and estimated the $I_{DLD}$. This study then explored the characteristics of $I_{HERI}$ that affect landslides under specific natural environmental factor and slope land use disturbance conditions. The weight analysis results indicate that of the slope land use disturbance factors, bare land density exerted the strongest effect. Of the natural environmental factors, slope coding exerted the strongest effect. Compared with earthquakes, rainfall exerted a stronger effect. Irrespective of whether landslides were induced by a single rainfall event, a single earthquake, or both rainfall and earthquakes, the grid ratio for landslides in the study area tended to increase with an increase in the degree of slope land use disturbance. When $I_{HERI}$ or $I_{DLD}$ was larger, the induced landslide ratio increased. Thus, when the interaction

between rainfall and earthquakes is considered, the impact of slope land use disturbance on landslide hazards should be noted. Moreover, regardless of whether single rainfall, single earthquake, or both rainfall and earthquake events were considered, the accuracy of each landslide potential model constructed using the RF method was above 83%, which is considered excellent. The AUC values in the assessment results for landslides induced by single rainfall, single earthquake, postrainfall earthquake, and postearthquake rainfall events were all above 0.94, indicating a higher risk of landslide induction by single rainfall and earthquake events. The forecasts had a high accuracy. With the exception of when the average rainfall in the study area was higher than that during the dual trigger events, the area ratio of medium–high-level landslides induced by the interaction between rainfall and earthquakes was large. According to the estimated landslide potential in this study, irrespective of whether the average $EAR \times I_{3\text{Rmax}}$ value of a single rainfall event is larger or the maximum PGA value of a single earthquake is larger, the area of medium–high-level landslides tends to increase.

**Author Contributions:** Conceptualization, C.-M.T., Y.-R.C. (Yie-Ruey Chen), C.-M.C., Y.-L.Y., Y.-R.C. (Yu-Ru Chen) and S.-C.H.; methodology, C.-M.T., Y.-R.C. (Yie-Ruey Chen) and C.-M.C.; investigation Y.-R.C. (Yu-Ru Chen) and S.-C.H.; writing—original draft preparation, C.-M.T. and Y.-R.C.; writing—review, C.-M.T., Y.-R.C. (Yie-Ruey Chen), C.-M.C., Y.-L.Y., Y.-R.C. (Yu-Ru Chen) and S.-C.H.; editing, C.-M.T. and Y.-R.C. (Yie-Ruey Chen); supervision, C.-M.T. All authors have read and agreed to the published version of the manuscript.

**Funding:** This work was supported in part by grants from the Taiwan National Science and Technology Council (NSTC 110-2625-M-309-001 and NSTC 111-2625-M-309-001). The authors wish to express their appreciation to Disaster Prevention Research Center, NCKU, for providing the technical support of ENVI.

**Institutional Review Board Statement:** Not applicable.

**Informed Consent Statement:** Not applicable.

**Data Availability Statement:** Not applicable.

**Conflicts of Interest:** The authors declare no conflict of interest.

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
