# Peer review of "Statistical Analysis of the Potential of Landslides Induced by Combination between Rainfall and Earthquakes"

_water, doi:10.3390/w14223691_

Round 1

Reviewer 1 Report

The topic and the case-study of the manuscript are interesting and challenging. The analysis of the potentially effects of the rainfall and earthquakes as trigger factors of landslide phenomena is of great scientific interest. Furthermore, I greatly appreciated the large amount of data considered in this study.

However, I have to point out many critical aspects in the manuscript.

The title could be misleading, the study concerns the analysis of the combined effect of the rainfall and earthquakes to trigger landslides. There are no correlations between rainfall and earthquakes. 

The paragraph "Research Methods" is a list of the methods used for the data analysis. Some methods are trivial and they do not require description (i.e. GIS), some others (i.e. RF algorithm) requires more details. 

There is an excessive use of tables and indexes to describe the data and some results. A focus to the critical parameters and indexes could make more readable the paper.

In particular, the paragraph 4.5 is quite difficult to follow. (for example the sentence: The out-of-bag error rate was used to estimate the error of the potential landslide assessment).

There are no comments about the limits of the proposed approach.

The paragraph "Conclusions" is a sequence of short sentences, this is not acceptable in a scientific paper! I strongly suggest to better emphasize the novelty of scientific results and implications with future works.

The figure captions could be more informative with the introduction of details on data and results displayed in the graphs.   

Finally, the English form requires a deep and accurate revision.

Author Response

Please see attached PDF file, many thanks.

Reviewer 2 Report

The Authors investigated the potential of a machine learning technique, based on the random forest data mining algorithm applied to satellite images, to evaluate the landslide susceptibility of two large areas, in Taiwan. The Authors have considered 6 natural environmental hazard factors (elevation, slope, aspect, distance from the river, geology, distance from the fault) and 1 disturbance factor related to slope utilization (green coverage, crop density, building density, road density, bare density).

To confirm the accuracy of the satellite images interpretation, 25 points were selected as checkpoints (interpretation grids). Then, the Authors defined two hazard indexes (IHERI related to earthquakes and rainfalls and IDLD related to degree of land disturbance) to explore their mutual relationship obtaining some results that they comment in the conclusions. 

The use of machine learning techniques for the study of landslide hazard of large areas is a promising tool in helping researchers and authorities to make rational hazard maps. However, this kind of analysis is a sort of “black box” and the Authors should be very careful in providing checks to validate their results. In this sense, I believe that the paper does not yet provide clear demonstrations of the validity of the presented results. In particular:

A.      It is not clear what the 25 points of the interpretation grids are (L239-240) – this aspect is the only way to validate the reliability of the analysis and should be carefully considered;

B.      It is not clear what is the basic grid of analysis (L322: 40 x 40m? L434: 2,339,322?), what is the relationship between the basic grid of analysis and the satellite image resolution (L219: 6x6 m, 8x8 m, 10 x 10 m) and what it implies with the detection of landslides; 

C.      It is not clear how the satellite images interpretation identifies a landslide and what kind of landslides have been identified by satellite image interpretation (what is the minimum extension of the landslides that have been detected by using satellite images interpretation? May be that only large landslides involving bedrock are included?)

D.      It is not clear how the hazard susceptibility map obtained by analysing satellite images have been validated with existing landslide map (Where is the control area? What kind and extent of landslides are included in the control area? How were these landslides mapped?) 

Also, I think the Introduction should include some more relevant papers on the issue you discussed. Among others, it is suggested to consider:

I.            F. Catani, D. Lagomarsino, S. Segoni, and V. Tofani (2013). Landslide susceptibility estimation by random forests technique: sensitivity and scaling issues. Nat. Hazards Earth Syst. Sci., 13, 2815–2831, 2013. doi:10.5194/nhess-13-2815-2013

II.            Goetz, J.N., Brenning, A., Petschko, H. & Leopold, P. (2015). Evaluating machine learning and statistical prediction techniques for landslide susceptibility modelling. Computers and Geosciences, vol. 81, pp. 1-11.

III.            Steger, S., Brenning, A., Bell, R., Petschko, H. and Glade, T. (2016). Exploring discrepancies between quantitative validation results and the geomorphic plausibility of statistical landslide susceptibility maps. Geomorphology, 262, pp. 8-23.

IV.            Khaled Taalab, Tao Cheng & Yang Zhang (2018). Mapping landslide susceptibility and types using Random Forest, Big Earth Data, 2:2, 159-178, DOI: 10.1080/20964471.2018.1472392

Moreover, specifically on the issue of rainfall and earthquake induced landslides, it is suggested to consider some more references:

V.            Jibson, R. W., Prentice, C. S., Borissoff, B. A., Rogozhin, E. A. & Langer, C. J. (1994). Some observations of landslides triggered by the 29 April 1991 Racha earthquake, Republic of Georgia. Bull. Seismol. Soc. Am. 84, 963–973 (1994).

VI.            Kefeer, D.K.; Manson, M.W. (1998). Regional distribution and characteristics of landslides generated by the earthquake. In The Loma Prieta, California, Earthquake of October 17, 1989-Landslides; U.S. Geological Survey Professional Paper 1551-C; Kefeer, D.K., Ed.; U.S. Geological Survey: Reston, VA, USA, 1998; pp. 7–32.

VII.            Ruggeri, P.; Fruzzetti, V.M.E.; Ferretti, A.; Scarpelli, G. (2020). Seismic and Rainfall Induced Displacements of an Existing Landslide: Findings from the Continuous Monitoring. Geosciences 2020, 10, 90. https://doi.org/10.3390/geosciences10030090

VIII.            Lacroix, P., Gavillon, T., Bouchant, C. et al. SAR and optical images correlation illuminates post-seismic landslide motion after the Mw 7.8 Gorkha earthquake (Nepal). Sci Rep 12, 6266 (2022). https://doi.org/10.1038/s41598-022-10016-2

 Furthermore, some aspects should be checked:

 1.       The title does not contain any reference to the statistical analysis or machine learning algorithm that has been used. Please, consider the hypothesis to focus the title, for example: “STATISTICAL analysis of the potential of landslides…” or “Analysis of the Potential of Landslides Induced by Interaction Between Rainfall and Earthquakes USING RANDOM FOREST ALGORITHM” or similar.

2. The landslide hazard factors described in section 4.1, do not match with the list indicated in lines 232-234. Please, make the list equal to the factors indicated in section 4.1, or explain the difference;

3.       Why the results of table 1 are given before the description of landslide hazard factors of section 4.1? Please, explain;

4.       L305 – Distance from the fault: please explain if this distance is related to the seismogenetic fault of the earthquake or it is related to the structural arrangement of the area (i.e. every fault is considered as damaging the rock formation);

5.       Table 7, Table 10: check the significant figures

6.       Table 12: check “Taining” – “Training”

Author Response

(The authors gave the same response as above.)

Round 2

Reviewer 2 Report

Dear Authors, I appreciated the replies to my comments and the revision of the paper. I recognize the paper analyses a large amount of data and can be a useful trace for future work on this field. However, one aspect that still needs to be added in the paper is the type of landslide you have considered: as you mentioned in your reply, you have considered DEBRIS FLOWS that “were the easiest and most reliable type to be identified on satellite images since vegetation was usually stripped off from the slopes”. Unfortunately, I did not find this aspect reported in the article while I think it should be clearly indicated both in the body of the manuscript and in the conclusion.

Also, for the future development of the work, I recommend to show a little area where the grid of analysis, the existing landslides and the results in term of landslide hazard are clearly visible. Currently the map of the entire region is not very informative because the reader is not able to adequately detect the hazard areas limit and the existing landslides perimeter, so the only available evidence is the percentage in the tables.

After adding the specification on the considered type of landslides I think the paper can be published in Water.

Author Response

Please see attached PDF file, many thanks
